# Anti-Hormonal Therapy in Breast Cancer and Its Effect on the Blood-Brain Barrier

**DOI:** 10.3390/cancers14205132

**Published:** 2022-10-19

**Authors:** Carolin J. Curtaz, Ludwig Kiesel, Patrick Meybohm, Achim Wöckel, Malgorzata Burek

**Affiliations:** 1Department of Gynecology and Obstetrics, University Hospital Würzburg, 97080 Würzburg, Germany; 2Department of Gynecology and Obstetrics, University Hospital of Münster, 48143 Münster, Germany; 3Department of Anaesthesiology, Intensive Care, Emergency and Pain Medicine, University Hospital Würzburg, 97080 Würzburg, Germany

**Keywords:** anti-hormonal therapy, brain-metastasis, blood-brain barrier, breast cancer

## Abstract

**Simple Summary:**

Anti-hormonal therapie regimes are well established in oncological treatments in breast cancer. In contrast there is limited knowledge of their effects on metastatic brain metastases in advanced breast cancer and their ability to cross the blood brain-barrier. In this review, we point out the usual antihormonal therapy options in the primary disease, but also in metastatic breast cancer. In addition, we explain the epidemiological facts of brain metastases, as well as the basics of the blood-brain barrier and how this is overcome by metastase. Last but not least, we deal with the known anti-hormonal therapy options and present clinical studies on their intracerebral effect, as well as the known basics of their blood-brain barrier penetration. Not all common anti-hormonal therapeutics are able to penetrate the CNS. It is therefore important for the treating oncologists to use substances that have been proven to cross the BBB, despite the limited data available. Aromataseinhibitors, especially letrozole, probably also tamoxifen, everolimus and CDK4/6 inhibitors, especially abemaciclib, appear to act intracerebrally by overcoming the blood-brain barrier. Nevertheless, further data must be obtained in basic research, but also health care research in relation to patients with brain metastases.

**Abstract:**

The molecular receptor status of breast cancer has implications for prognosis and long-term metastasis. Although metastatic luminal B-like, hormone-receptor-positive, HER2−negative, breast cancer causes brain metastases less frequently than other subtypes, though tumor metastases in the brain are increasingly being detected of this patient group. Despite the many years of tried and tested use of a wide variety of anti-hormonal therapeutic agents, there is insufficient data on their intracerebral effectiveness and their ability to cross the blood-brain barrier. In this review, we therefore summarize the current state of knowledge on anti-hormonal therapy and its intracerebral impact and effects on the blood-brain barrier in breast cancer.

## 1. Introduction

The average risk of developing breast cancer is around 13% for the general population and, with around 69,000 initial diagnoses, shows the highest tumor entity in woman in Germany, but also all over the western world [1]. In addition, breast cancer causes the most tumor-associated deaths of women from industrialized countries [2,3]. Tumor metastasis plays a role in mortality and overall survival (OS). Through differential early detection screening measures, but also advanced preventive examinations, mortality has been significantly reduced in recent years [4]. Despite all this, many patients die prematurely due to a pronounced tumor affliction.

Around 15–20% of breast cancer patients develop brain metastasis (BM) during the course of their disease [5]. This correlates with a very poor prognosis but also with a major lack of quality of life, as cognitive and sensory functions are impaired [6]. The incidence of BM is raising in recent years, probably due to better OS resulting from effective palliative therapy regimes and more frequent use of MR diagnostics [7,8,9].

The central nervous system (CNS) is protected by a natural barrier called the blood-brain barrier (BBB), which consists of specialized brain microvascular endothelial cells, including other components of the neurovascular unit like astrocytes, pericytes, neurons, microglia, and extracellular matrix [10].

The key event in the cerebral spread of metastases involves the crossing of this barrier by tumor cells. However, the mechanisms behind this invasion are not well understood. In more recent studies, scientists are trying to find out whether tumor-cell derived molecular serum markers, such as cytokines or exosomes and their content [11,12,13], could play a role in tumor cells overcoming the BBB. In addition, oncological therapeutic medications are increasingly becoming the focus of interest. The main clinical question is whether BM can be treated with this medication and whether the BBB is permeable to it. On the other hand, these therapeutics might also interfere with the BBB and increase permeability, thus supporting the spread of breast cancer into the CNS.

Hormone-positive breast cancer presents the largest subgroup with the best OS and mortality [14,15]. Unfortunately, this subtype of breast cancer also develops metastatic sides including BM. Although mainly HER2−positive and triple-negative breast cancer (TNBC) develop BM, an increasing number of patients with luminal A-like and luminal B-like carcinomas also suffer from BM [16,17]. In the last decades, in the adjuvant but also in the palliative state of breast cancer disease, mainly anti-hormonal mono-therapy regimes have been used in the treatment regimes, which in frequent cases have shown their limitations in tumor containment, since anti-hormonal resistance has been developed by the tumor cells. With the introduction of CDK 4/6 inhibitors [16] in combination with already established anti-hormonal medications in 2016, a very effective new group of drugs was established which, for the first time in oncological breast cancer therapy, showed not only a longer progression free survival (PFS), but also a longer OS [18,19]. Another class of drugs for advanced metastatic hormone-positive breast cancer with PIK3 modification-PIK3-inhibitors [20] have recently been brought to market.

While mono-anti-hormonal therapies are well established in oncological treatments, there is limited basic research knowledge of their effect on the BBB. The effect on their impact on BM is based on clinical research and experimental animal models. There are also limitations in basic research for the newly introduced medications (CDK4/6 inhibitors and PIK3 inhibitors).

This review summarizes published clinical and basic data on established current anti-hormone therapy medication in breast cancer and describes their impact on the BBB.

## 2. Breast Cancer

### 2.1. Molecular Characteristics

Breast cancer is differentiated into histological subtypes according to the St. Gallen Classification. Based on the analysis of biological markers in the primary tumor, there are four subtypes based on estrogen receptor (ER)-status, progesterone receptor (PR)-status (PR), human epidermal growth factor receptor 2 (HER2) and proliferation marker Ki67, together with tumor size, histological grade and lymph node engagement [21,22]. The four-type classification of the different types of breast cancer into luminal A-like (ER + and/or PR+, Ki67 low and HER2−), luminal B-like (ER+ and/or PR+, Ki67 high and/or HER2+), HER2 positive (ER+/−, PR+/− and HER2+) and triple-negative (ER−, PR− and HER2−). The HER2−positive group consists of breast cancer, which shows the molecular pattern of HER2 positive, no matter which HR status is given. If the HR-status is ER positive and PR positive it is also called triple positive (ER+, PR+ and HER2+). Its prevalence among the HER2−positive subgroup is about 50%, although their ER expression may be at lower levels [23]. This classification gives a prediction of disease characteristics, recurrence pattern and disease-free survival, but also helps to provide therapy guidelines [24].

About 80–85% of breast cancers are histologically hormone receptors-positive (HR-positive; ER- and PR-positive) and belong to the luminal A-like or luminal B-like carcinoma type. It is difficult to predict the exact timing of breast cancer recurrence, as the period ranges from months to decades after surgery [25,26].

Luminal A-like breast cancer is associated with the lowest rates of recurrence compared to all other subtypes, and together luminal A-like and luminal B-like show comparatively later recurrence in epidemiologic studies [27]. Van Maaaren and colleagues could demonstrate that about 4.7% of patients suffered a local recurrence within 10 years, 3% had a regional recurrence and in 15% had distant metastases. Other epidemiological studies show similar results [26,28].

Subdivided into molecular subtypes, distant metastases within 10 years occurred most frequently in patients with initially HER2−positive breast cancer (25.6%), TNBC (23.2%) and 20% of patients with luminal B-like disease and 9.5% of patients with luminal A-like disease [27].

### 2.2. Current Therapies of Primary Hormone-Positive Breast Cancer

Typically, treatment of breast cancer includes surgery, usually radiation, and systemic chemotherapy (adjuvant, neoadjuvant, or even both) for subtypes of higher risk or anti-hormone therapy for subtypes of endocrine origin [24]. Since most breast cancers belong to the ER-positive breast cancer subtype, this histopathological group is subdivided into luminal A-like and luminal B-like. The luminal A-like subtype is characterized by a low proliferation rate (indicated by a Ki-67 of less than 20%) and thus has a more favorable prognosis compared to the luminal B-like subtype [29]. The grouping has extraordinary implications for the therapeutic treatment of the patient. Luminal A-like breast cancer is treated by surgery, most often with radiation, while systemic therapy is limited to endocrine therapy for at least 5 years after the initial treatment. Endocrine therapy differs between premenopausal and postmenopausal patients. Premenopausal women are recommended therapy with tamoxifen (a selective estrogen receptor modulator, SERM) or tamoxifen plus ovarian suppression for 2–5 years (for patients at higher risk of recurrence) or aromatase inhibitor (AI) plus ovarian suppression for 5 years (for patients with increased risk of recurrence) [30]. According to the TEXT/Soft study, the sequential use of AIs and tamoxifen is recommended for postmenopausal women for the first 5 years (tamoxifen (2–3 years) followed by AI or AI (2–3 years) followed by tamoxifen up to a total of 5 years) [31,32]. Certain general recommendations are given for hormonal therapy [33]. The adjuvant endocrine therapy is divided into initial therapy (years 1–5) and extended adjuvant therapy (EAT, years 6–10+). The standard duration of adjuvant therapy is 5 years. The extended duration of therapy is evaluated after an individual benefit-risk assessment and discussed with the patient. The duration, choice and sequence of AI or Tamoxifen depend mainly on individual menopausal status, tolerability and risk of recurrence. Switching to another endocrine therapy (tamoxifen or AI) is better than stopping therapy. Beginning with AI is particularly recommended in the case of lobular carcinoma and/or a significantly increased risk of recurrence. There is no sufficiently validated biomarker for early versus late recurrence [34].

In contrast to luminal A-like breast cancer, the luminal B-like subtype is characterized by a high rate of proliferation and/or a high histological grade. It is also associated with a higher cumulative incidence of distant metastases and an increased risk of recurrence [30]. Therefore, it is recommended to treat these patients by surgery, mostly radiation (if not treated by mastectomy) plus systemic chemotherapy, followed by endocrine therapy for at least 5 years [30,35].

Until recently, the systemic chemotherapy was used after surgery in the so-called adjuvant setting. Recently, however, the use of neoadjuvant chemotherapy (NAC) has become increasingly preferred. It is the intention of NAC to expand surgical options and avoid mastectomies for patients and also improve their survival [36]. Chemotherapy contains systemic components that do not explicitly attack the hormone-sensitive components of the tumor, but have a generally inhibitory and cytotoxic effects. Therefore, for luminal B-like subtypes, conventionally dosed antracycline-taxan-based chemotherapy (q3w) is often used, while dose-dense chemotherapy (including weekly schedules) followed by endocrine-based therapy is recommended [35]. However, there are also promising approaches for new therapy options to reduce the risk of recurrence in the postadjuvant and postneoadjuvant setting. Just recently, in May 2022, the approval for abemaciclib, a CDK 4/6 inhibitor for adjuvant therapy with a high risk of recurrence, was extended [37]. Therefore, abemaciclib is recommended for 2 years for patients with an increased risk of recurrence and characteristics that meet the study criteria of the MonarchE study. Adjuvant maintenance therapy with PARP inhibitor olaparib for 1 year is also recommended for patients with a BRCA1- or BRCA2-mutated breast cancer (gBRCA1/2) [35].

In addition, in the case of high risk of recurrence, Extended Adjuvant Endocrine Therapy (EAT) (years 6–10) is increasingly recommended. Factors indicating a clinical benefit of EAT are adjuvant tamoxifen therapy alone, post-chemotherapy status (indicative of high risk), positive lymph node status and/or T2/T3 tumors, increased risk of recurrence based on immunohistochemical criteria or based on multi-gene expression assays, and high CTS5-score (online model for clinicians to predict late distant metastasis in women with ER-positive breast cancer) [35]. Long-term therapy of at least 5 years, and even up to 10 years in high-risk groups, is therefore an essential part of adjuvant breast cancer tumor therapy. Next to the medication side effects, such as sleep disorders, hot flashes, pain and anxiety, as well as a reduced quality of life, this has an impact on therapy adherence [38]. However, the CNS side effects also show that these substances must affect the brain and therefore can pass through or affect the BBB.

### 2.3. Current Therapies of Hormone Positive Metastatic Breast Cancer

Endocrine therapy is the gold standard treatment in patients with metastatic breast cancer and HR-positive (or unknown) status. There is one exception, namely the threat of organ failure. However, it must be kept in mind that HR status can also change over the course of the disease. Therefore, if possible, a histology of the recurrent site should be performed [35]. General treatment considerations include that for all lines of treatment, treatment options should take into account prior endocrine therapies, age and comorbidities, and regulatory status. Premenopausal patients treated with GnRH analogues or after ovariectomy can be treated like postmenopausal patients [35]. In pharmacology, analogs are used to describe an active ingredient that, by binding to a receptor, develops the same effect as the endogenous ligand. GnRH analogues bind to the GnRH receptors in the pituitary gland accordingly. The mechanism of action lies in a complete down-regulation of the pituitary receptors, which, after an initial brief increase in gonadotropin and thus sex hormone secretion, leads to a complete inhibition of sex hormone secretion. They therefore have an anti-hormonal effect and in the case of breast cancer they have an anti-oestrogenic effect [39].

In general, the endocrine therapy for premenopausal patients with HER2−negative metastatic breast cancer includes ovarian suppression with GnRH analogues or, more rarely, with tamoxifen alone [40,41,42]. Tamoxifen blocks the cytoplasmic ER through competitive inhibition, resulting in reduced cell division activity in estrogen-dependent tissues [43].

In its last statement from 2022, the AGO recommends therapy with GnRHa + Fulvestrant + CDK4/6i [35] with the highest recommendation grade or using GnRHa + AI + Ribociclib based on results of the MONALEESA-7 trial [35,44,45]. Treatment regimes with GnRHa + AI + Palbociclib/Abemaciclib, GnRHa + tamoxifen, GnRHa + AI (first + second line) or GnRHa + fulvestrant have also been recommended [35]. The latest published ESMO (European Society for Medical Oncology) Clinical Practice Guideline from October 2021 recommends in the first-line treatment CDK4/6i combined with ET as the standard-of-care for ER-positive, HER2−negative MBC, with improved PFS and OS and a good toxicity profile seen in several trials [42]. The ESMO recommends for patients who did not relapse on an AI, or within 12 months of stopping adjuvant AI, a CDK4/6i in combination with an AI, with no clear advantage of fulvestrant seen in a phase II study [42,46].

Endocrine-based treatment options for postmenopausal patients with HER2−negative metastatic breast cancer include recommendations similar to those for premenopausal patients. However, monotherapy regimes such as fulvestrant, AI or tamoxifen alone are also possible [35,42], but should be reserved for the small group of patients with comorbidities or a PS that precludes the use of CDK4/6i combinations recommends the ESMO [42]. In its last statement from 2022, the AGO recommends a therapy regimen with a CDK4/6i (abemaciclib, palbociclib, ribociclib) plus non-steroidal AI e.g., exemstane or/plus fulvestrant with the highest recommendation grade [35]. The treatment with alpelisib plus fulvestrant in PIK3CA-mutated patients or everolimus, an mTOR inhibitor, plus exemestane or everolimus plus tamoxifen are also recommended as alternative therapy regimens [35]. Pure endocrine monotherapy is not indicated in patients in whom rapid remission is required to avert pronounced symptoms of the affected organ [40]. The ESMO recommendations are based on first-line treatment, second-line and beyond second-line treatment. Next to the first-line treatment (mentioned above) as second line treatment the ESMO recommends that chemotherapy versus further endocrine-based therapy should be based on disease aggressiveness, extent and organ function, and consider the associated toxicity profile. Next to that, alpelisib plus fulvestrant is mentioned as a treatment option for patients with PIK3CA-mutant tumors, prior exposure to an AI and appropriate HbA1c levels. Everolimus plus exemestane is recommended as an option since it significantly prolongs PFS. Tamoxifen or fulvestrant can also be combined with everolimus. It is also recommended that PARP inhibitor monotherapy (olaparib or talazoparib) should be considered for patients with germline pathogenic BRCA1/2 mutations as an option for those with somatic pathogenic or likely pathogenic BRCA1/2 or germline PALB2 mutations. The ESMO advises at least two lines of endocrine-based therapy are preferred before moving to a systemic chemotherapy, as for patients with imminent organ failure chemotherapy is a preferred option [42].

A problem with the use of endocrine therapy regimens is the recurrence of endocrine resistance progression of the metastatic disease. There is primary endocrine resistance or secondary resistance. Primary endocrine resistance includes relapse within 2 years of adjuvant endocrine treatment (ETx) or a progressive disease within first 6 months of first-line ETx in metastatic breast cancer. Secondary endocrine resistance is defined as recurrence during adjuvant ETx but after the first 2 years of ETx or as recurrence within 12 months of completion of adjuvant ETx, or when progressive disease is diagnosed 6 months after initiation of ET in metastatic breast cancer [47].

An advantage of endocrine therapy is that it is usually well tolerated by patients and most therapeutics are oral therapies that are easy to take. Therefore, in addition to endocrine resistance, a visceral crisis is also an indication for a change in therapy regimen to chemotherapy regimen with or without targeted drugs. A visceral crisis is defined as severe organ dysfunction. This is determined by signs and symptoms and laboratory tests for rapid disease progression. A visceral crisis is not only defined by the presence of visceral metastases, but implies an important organ compromise that leads to the clinical indication for the fastest effective therapy [47].

## 3. Brain Metastases in Breast Cancer (BMBC)

### 3.1. Epidemiology

Breast cancer patients with primary tumor characteristics such as HR-negative status (basal-like cell type/triple-negative), high grade, high Ki-67 index, HER2 overexpression, molecular subtype (luminal B-like, HER2 positive, triple-negative) have a higher risk of developing BMBC [17,48,49,50,51]. About 30–50% of patients with metastatic breast cancer (MBC) develop BM in the course of their disease [52].

Overall, BM tend to have a histological ER-negative status and overexpression of HER2 and/or EGFR [53,54,55,56]. Only in a small percentage of patients with HR-positive breast cancer (0.1–0.2% for luminal A-like, and 0.6–3.3% for luminal B-like [47,54,57]) will develop BM.

The Brain Metastases in Breast Cancer Network Germany Register [17,58,59], a register study that presents prospective and retrospective data on the course of disease in patients with BM from breast cancer, showed that the median life span after diagnosis of a cerebral metastasis is only 7 months. The 1-year survival rate is 37%. In disease caused by meningiosis carcinomatosa, the median survival time is only 3 months [17,58]. Kuksis et al. analyzed the incidence of BM in a meta-analysis of articles published from January 2000 to January 2020. The pooled cumulative incidence of BM was 31% for the HER2+ subgroup, 32% for the triple- negative subgroup and 15% in patients with HR+/HER2− MBC. He also analyzed the corresponding incidences per patient-year: These were 0.13 for the HER2+ subgroup, 0.13 for the triple-negative subgroup, and only 0.05 for patients with HR+/HER2− MBC [53]. In analysis of real OS data from 3500 patients with BMBC, Darlix et al. showed that at a median follow-up of 30-month after the diagnosis of CNS metastases, 74% of all patients had died. The median OS after diagnosis of CNS metastases was 7.9 months. The 6-, 12-, 24- and 48-month survival rates were 56.3%, 37.7%, 22.1% and 8.0%, respectively [60]. The sub-analyses regarding receptor status showed that median OS was 7.1 months for HER2−/HR+ (Luminal), 18.9 months for HER2+/HR+, 13.1 months for HER2+/HR− and 4.4 months for triple-negative tumors [60]. With an OS of only 7.1 months, LuminalBMBC show the second lowest OS rate after the OS triple-negative BMBC in this analysis.

The question arises if one reason for this low OS of Luminal BMBC and triple-negative BMBC could be that there are more clinically approved HER2+ drugs, which can penetrate the brain and therefore treat BM than for Luminal BC or TNBC. Do commonly used therapeutics, as for this type of BMBC, anti-hormone medications including AIs, SERMs and CDK4/6i, overcome the BBB and could therefore treat BM? Do those medications affect the BBB?

### 3.2. Overcoming the Blood-Brain-Barrier

The formation of BM is based on tumor cells crossing the so-called BBB. The BBB is a highly specific permeable membrane consisting of endothelial cells, pericytes and astrocytes. With tight junctions (TJs) and adherens junctions (AJs), the endothelial cells are closely connected, so that paracellular exchange of substances and fluids can be largely prevented [61], although limited transcellular transport occurs [62]. In the case of a BM, this barrier property is disturbed (see Figure 1) [63].

However, it is still unclear how tumor cells exactly penetrate the BBB. The diapedesis of the BBB, also called extravasation process, of tumor cells seems to show organ-specific elements [64] and specific tumor microenvironment in BM [65]. It has been shown that tumor cells need significantly longer to invade the brain than other organs [66]. The question arises if and how cancer cells in the long term alter the blood-brain barrier and therefore affect the ability of drugs to reach the brain. Functionally, the question arises as to whether tumor cells leave the endothelium intact during diapedesis or whether they destroy the vessel wall. Overall, there is no unanimous opinion how this happens [67,68]. Breast cancer cells were shown to migrate through the BBB at sites of vessel wall discontinuity, but no endothelial apoptosis or hypoxia was observed [64]. In principle, breast cancer cells could benefit from the protection of the BBB from anti-tumor immune cells, but also from chemotherapeutic agents or specific immunotherapeutic agents. It is controversially discussed whether the vascular network of the metastases shows an altered permeability. It is also unclear whether the blood-tumor barrier remains intact in small metastases and only changes its integrity when larger lesions are formed [69]. Other authors, however, argue that the blood-tumor barrier has increased permeability in breast cancer metastases. This also does not correlate with the size of the lesion. However, it is assumed that the barrier remains sufficiently intact for drug delivery [70].

**Figure 1 cancers-14-05132-f001:**
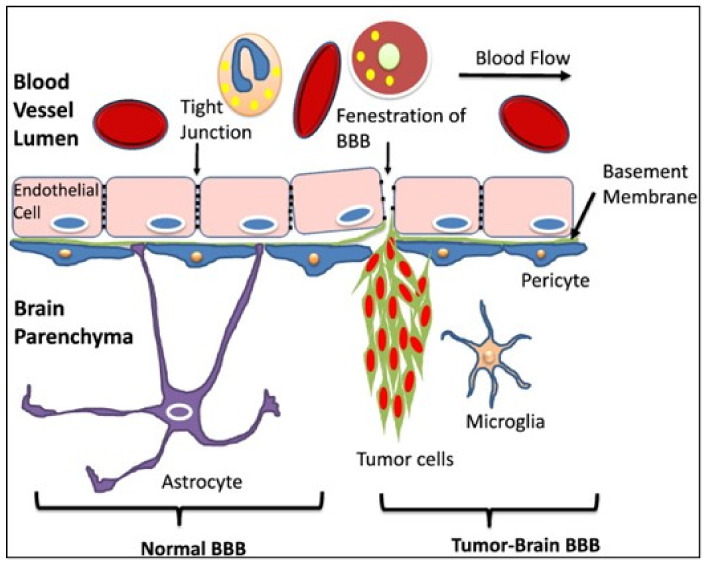
Structure of the BBB in the healthy brain and in the tumor brain [71].

There are different transport routes through the BBB. Passive diffusion is one way of crossing the BBB. A large number of different fat-soluble molecules can therefore passively diffuse through the BBB and thus reach the brain. This mechanism has no upper limit and depends on whether the drug penetrates the cell membrane [72]. Low molecular weight and a high lipid solubility favor transport by this type of mechanism. In addition, lipid solubility also favors uptake by peripheral tissues, but it may also reduce the amount of drug delivered to the brain [73].

The second way to cross the BBB involves active efflux transporters. They are composed of multiple ATP-binding cassette (ABC) proteins expressed on the luminal blood-facing endothelial plasma membrane of the BBB. These ATP-driven efflux pumps for many different types of metabolites significantly restrict the permeability of the BBB for toxins and therapeutics. The drug-resistant properties of the CNS are attributed to the high expression of ABC-transporters [74]. These efflux pumps include P-glycoprotein (P-gp) [75,76], Organic Anion Transporting Polypeptides (OATP) [77,78] or Breast Cancer Resistance Protein (BCRP) [76,79]. Some of these efflux systems also appear to have overlapping substrate affinities, such as P-gp and BCRP [80]. They can pump various molecules and substrates from the endothelial cells back into the bloodstream [81].

Since the BBB isolates the brain so sufficiently that even it restricts the diffusion of many essential nutrients, including glucose and amino acids, another transport pathway called carrier-mediated transport (CMT) exists for this necessary exchange. CMTs are encoded by the transporter genes called Solute Carrier (SLC). This gene family includes more than 300 transporter genes. They encode membrane-bound proteins that enable the transport of a variety of substrates, such as amino acids, fatty acids, hormones, carbohydrates, monocarboxylic acids and many more across biological membranes [82].

Larger peptides and proteins are constrained by their peptide bonds, preventing them from using the amino acid-CMT systems to cross the BBB. Hence, there is receptor-mediated transport (RMT) and the adsorptive-mediated transcytosis (AMT) that allow these molecules to enter the brain through endocytotic mechanisms in a process called transcytosis [83,84]. Transport by RMT occurs by binding of macromolecules to ligand specific receptors on the cell surface, which in turn triggers an endocytic event. The receptor and its bound ligand fuse result in a caveolae that pinches off to form a vesicle. Both the ligand and the receptors are taken up and passed through the cytoplasm to be expelled on the opposite side of the cell via exocytosis [61]. When transported via AMT, positively charged large molecules interact with specific binding sites on the cell surface. Then endocytosis and transcytosis are induced [85].


*The CNS is respected as an immune-privileged site. The reason for this is that, compared to other tissues, very few neutrophils infiltrate the brain and this also represents a tightly regulated interaction between immune cells and BBB [86]. But inflammatory and other pathological conditions *e.g.,* tumor metastases, can disrupt the TJs between endothelial cells. Cytokines and other pro-inflammatory agents may play a role in BBB specific permeability loss [12]*


### 3.3. Therapy Options of Brain Metastases in Breast Cancer

Recently, there have been significant advances in the development of cancer therapeutics, including molecularly targeted drugs and novel immunotherapies. Both targeted therapies and immunotherapeutics have demonstrated unequivocal efficacy in tumors at the primary site, but delivery of these therapeutics to many metastatic sites in the brain across an intact BBB remains a challenge.

To date, the gold standard treatment for BMBC is by surgery, radiosurgery and radiation [17,87]. These locoregional treatment options are the only ones currently showing a significant positive impact on the survival of patients with BM [88,89,90]. Additionally, local therapy surgery, stereotactic radiosurgery (single session) (=SRS), and fractionated stereotactic radiotherapy (=FSRT) depends on localization, size, number of metastases, previous therapy, Karnofsky-Performance-Scale and the prognosis of the patient [35,91]. Whole brain radiotherapy (WBRT) in addition to SRS/FSRT improves intracranial control of BM, but does not improve functional independence duration and OS [91,92]. A problem with this clinical data, however, appears to be that there is no good quality published data regarding brain surgery versus no surgery in patients with recorded brain metastases from breast cancer, and those data and these data are based solely on surgery versus radiosurgery at BMBC patients. A problem with the treatment with WBRT is that neurocognitive function is impaired, which can significantly reduce the patient’s quality of life. For this reason, the treatment with SRS/FSRT is preferred in the case of a limited number of BM [35,91,93,94]. The definition of an oligo-metastasis or a limited tumor volume is based on the description of ≤4 BM or a cumulative tumor volume <15 mL with 5–10 BM [30].

After the primary surgery, the question arises, will the patient benefit from postoperative radiotherapy? Analyses show that single/solitary BM (resection cavity < 5 cm) treated by SRS vs. WBRT do not differ in OS. But oligo-BM treated by SRS (surgical cavity and unresected metastases) versus WBRT also show no difference in OS [95,96,97,98]. Since patients often suffer from impairment of neurocognitive functions, symptomatic treatments of BM are recommended. This includes the use of anti-convulsants, but only if seizure symptoms occur [90,99]. Glucocorticoids should only be used if there are symptoms and/or mass effect [90,100,101,102]. However, for patients with bad prognosis and reduced general physical conditions, the best possible supportive treatment must be discussed and can be an option [35,90]. Besides the local treatment of BM, the systemic therapy of those remains a challenge. In its proposal from 2022, the AGO urgently recommends interdisciplinary treatment planning and a recommendation to continue the current systemic therapy if it is a first diagnosis of BM and stable extracranial disease [35]. Additionally, the ESMO recommends that multimodal treatment of BMs should be based on a careful individual assessment of the various contributions of surgery, radiation oncology, and medicine oncology [87].

However, the choice of therapeutics also poses a challenge in the treatment of BM. Not all cerebral metastases can be resected, so the histological subtype is not always clear and sometimes has to be derived solely from extracranial metastases. Hulsbergen and collegues could demonstrate that there is a significant percentage of HR+ patients, (22.8%), for whom a switch of subtype occurred mostly based HER2 [103]. Other authors were also able to present similar results and postulated that receptor discordance between the primary tumor and BCBM is therefore frequent. It also impairs survival in the event of receptor loss and represents a missed opportunity for the use of effective therapies when receptors are gained [104,105].

Another question is if luminal breast cancer cells metastasizing to the brain still retain the ability to respond to the same anti-hormonal drugs as the original tumor. This question cannot be answered conclusively. As with non-cerebral metastases, the endocrine resistant certainly plays a role [106], but also the not finally clarified changes in the permeability of the BBB in cerebral metastases (referring to Section 3.2).

So far, there is only treatment recommendation for Her2+ BMBC, as there is no scientific basis for any other subtypes based on study results [35]. This is because patients with cerebral metastases have almost always been excluded from participating in clinical trials if they have BM.

## 4. Clinical Trials and Drug Approval for Bmbc

Since patients with cerebral metastases were often not primarily included in clinical trials, the sub-analyses of patients who developed BM during the use of the therapy were carried out using only available data from the trials.

### 4.1. HER2+ BMBC

For HER2+ BMBC there exists more clinical data in comparison to all other subtypes. One of the earliest trials to discuss the benefit of HER2 medications is the Landscape trial [107]. In this trial, lapatinib and capecitabine were tested in patients with HER2+ breast cancer and BM, but without the use of pre-medication. The study was designed as a single-arm phase II study and included 45 patients. The primary endpoint was defined as proportion of patients with an objective CNS response, meaning a volumetric reduction in CNS lesions of 50% or more. Overall, 66% of the patients had a partial remission [107]. However, the data are based on only a small study population.

The so-called CEREBEL-study was initiated to compare the incidence of CNS metastases as the first site of relapse in patients with HER2+ MBC receiving lapatinib-capecitabine or trastuzumab-capecitabine. The study showed a screening failure due to asysmptomatic BM at study enrollment. Therefore, the CEREBEL was inconclusive for the primary endpoint, and no difference was found between lapatinb-capecitabine and trastuzumab-capecitabine for the incidence of CNS metastases [108].

The results of the phase III trial CLEOPATRA by Swain et al. analyzed the long-term recurrence of BM. The authors found a longer time to development of CNS metastases in patients treated with pertuzumab-trastuzumab-docetaxel than in patient treated with trastuzumab-docetaxel [109].

A retrospective, exploratory analysis in the EMILIA-study was performed by Krop et al. [110]. There, the incidence of CNS metastases after treatment with trastuzumab emtansine (T-DM1) versus capecitabine-lapatinib, and the treatment efficacy in patients with pre-existing CNS metastases were analyzed. CNS progression was shown to be similar for T-DM1 and capecitabine-lapatinib in patients with HER2+ advanced breast cancer. However, in patients with treated, asymptomatic CNS metastases at baseline, T-DM1 was associated with significantly improved OS [110].

A notable clinical study investigating HER2+ patients with BM is the HER2CLIMB trial [111]. In this study, the intracranial efficacy and survival of patients treated with tucatinib plus trastuzumab and capecitabine were observed. The remarkable thing about this study is that patients with previous therapies were included for the first time and that the study was the only one that dealt with HER2+ patients with BM and that not only a subgroup analysis was carried out. In HER2+ breast cancer patients with BMs, the addition of tucatinib to trastuzumab and capecitabine doubled the objected-response rate, reduced the risk of intracranial progression or death by two-thirds, and reduced the risk of death by almost half [111]. These results of the study are also reflected in the last AGO recommendation on cerebral metastases in HER2+ BM patients [35]. Therefore, the use of tucatinib with trastuzumab and capecitabine is strongly recommended for these patients.

The first data the efficacy of the antibody-drug conjugate (ADC) trastuzumab deruxtecan (T-DXd) in patients with HER2+ BM are now available. A sub-analysis of the data from the Breast01 trial has been published [112]. The Breast01 trial is an ongoing two-part, multi-center, open-label, phase 2 trial of T-DXd. Twenty-four patients were enrolled and published data showed that the pattern of disease progression was similar in patients with and without BM, with 8 of 24 patients progressing (33%). Overall, first data showed that T-DXd had strong clinical activity both in the overall population of patients with HER2+ MBC and in the subgroup of patients with BMs [112]. Another study was published in August 2022 on T-DXd and BM. The TUXEDO-1 trial is a prospective study reporting the activity of T-DXd in patients with active BM, in which 15 patients were included [113]. A response rate of 73.3% and a progression free survival of 14 months could be demonstrated. These data appear very promising as these data are probably the longest reported period of any prospective study related to PFS, but they are limited to a small population [113].

In addition to ADCs, newly introduced irreversible pan-HER tyrosine kinase inhibitor (TKI) such as neratinib are also showing efficacy in HER2+ MBC. The NALA study, a randomized, active-controlled, phase III trial, compared neratinib plus capecitabine to lapatinib, a reversible dual TKI, plus capecitabine in patients with centrally confirmed HER2+ MBC. Included patients had more than 2 prior HER2−directed MBC regimens. Neratinib plus capecitabine showed a progression-free survival of 8.8 months compared to 6.6 months with the treatment regimen of lapatinib and capecitabine [114].

### 4.2. TNBC BMBC

Until recently, treatment options for patients with metastatic TNBC have been limited due to the lack of specific drugs. In the phase 3 ASCENT study, 529 patients with metastatic TNBC, who were refractory or relapsed after at least two prior chemotherapy regimens were randomized 1:1 to receive sacituzumab-govitecan (SG), an antibody-drug conjugate composed of an anti-Trop-2 antibody coupled to SN-38, or therapy of physician’s choice (TPC) [115]. A subgroup analyses of patients with BM from the ASCENT study showed that total of 61 of 529 (12%) enrolled patients had stable BM at screening and these patients were randomized to SG (*n* = 32) or TPC (*n* = 29). Median PFS was 2.8 months for SG versus 1.6 months for TPC, while median OS was 6.8 months SG versus 7.5 months for TPC [116]. The authors conclude that data interpretation in this population with poor prognosis is limited by the small sample size.

### 4.3. Luminal BMBC

Few clinical data are available for the luminal BMBC subtype. Only the CDK 4/6 inhibitor abemaciclib was tested in a specifically designed phase II study for patients with HR+ HER2− BM with cerebral metastasis [117]. However, the study did not meet its primary endpoint. Abemaciclib was associated with an intracranial clinical benefit rate (iCBR) of 24% in patients with heavily pretreated HR+, HER2− MBC. Abemaciclib has also been shown to reach therapeutic concentrations in BM tissue far exceeding those required for CDK4 and CDK6 inhibition. However, the authors of the study also believe that further studies are warranted, including the elevation of novel abemaciclib-based combinations [117,118].

### 4.4. Triple-Positive BMBC

The triple-positive subgroup (HER2+/ER+/PR+) with BM shows the best prognosis among patients with BM [58,119,120], compared to HER2 positive, hormone receptor-negative and HER2 negative BC [58], even though mostly the treatment is initially based on HER2 based medications. In order to better investigate this subgroup and to observe the use of endocrine substances in addition to the use of HER2 drugs, a single-arm study on the effectiveness of the combination of palbociclib, trastuzumab and lapatinib with fulvestrant in ER+/HER2+ BCBM (NCT04334330) was recently initiated in China [121]. Another study is currently testing tucatinib, abemaciclib and trastuzumab with specific secondary CNS endpoints (NCT03846583) [118,121]. Further analysis is urgently needed to evaluate the role of modern endocrine treatment concepts for this patient cohort.

## 5. Anti-Hormonal Therapy Regimensand Their Impact on the BBB

Elwood Jensen first described the ER in 1958 and showed that there are receptor-positive and receptor-negative tumors [122]. Almost at the same time, the anti-estrogen tamoxifen, originally used in contraception and reproductive research [123], was first developed by the English woman Marsha Cole as a hormonal form of therapy against breast cancer [124]. This, together with AIs, demonstrated a new mode of action: a drug that targets breast cancer and does not damage healthy tissue.

In the following we explain the current state of knowledge on the effects of common anti-hormonal drugs on BM and the BBB permeability in breast cancer (see Table 1) and their side effects only in relation to psychiatric disorders, the nervous system and the eyes.

### 5.1. Tamoxifen

The anti-estrogen tamoxifen acts via competition at the endogenous ER molecules and was first synthesized in 1966 [125] and has been used in the therapy of MBC since the 1970s [126].

The substance, which was introduced in 1973 as Nolvadex^®^ by the pharmaceutical company Zeneca (then ICI Pharma), first in the UK and in 1976 in the rest of Europe [127]. Tamoxifen was the first SERM because it has an anti-estrogenic effect on the breast, e.g., it has a partial estrogenic effect in the area of the bone and the endometrium [128,129].

The side effects of tamoxifen, such as psychiatric illnesses, e.g., depression is very common [130,131]. For diseases of the CNS such as drowsiness and headache, sensory disturbances (including paraesthesia and dysgeusia) are commonly described [132]. For diseases of the eyes, often reversible visual disturbances due to cataracts, corneal opacities (rare) and/or retinopathies (the risk of cataracts increases with the duration of the tamoxifen intake) occur. Rarely, optic neuropathy and optic neuritis occur in patients treated with tamoxifen (blindness has occurred in a small number of patients) [133,134].

Animal models have shown that tamoxifen can cross the BBB [135,136], so it could affect processes within the CNS that can be described as side effects. However, it is still unclear whether tamoxifen can inhibit the formation of BM and also suppress its growth and whether this is the reason why luminal like breast cancer develop less BM. There is too little clinical data to verify such a hypothesis. To date, there have been several clinical case reports of patients treated with tamoxifen who achieved remission [137,138,139].

### 5.2. Aromatase Inhibitor

Like the development of tamoxifen, the discovery of AIs as breast cancer therapeutics was rather accidental. The primary goal was to mimic surgical adrenalectomy and thus achieve an anti-tumor effect in postmenopausal women by lowering plasma estrogen levels [140]. In 1974, Thompson and Siiteri were able to demonstrate the mechanism underlying this phenomenon. This was achieved inducing the inhibition of aromatization by the substance aminoglutethimide [141], a so-called AI. Thus, the primary goal of chemical adrenalectomy was abandoned and aminoglutethimide, which belongs to the “non-steroidal” AIs, was born as a first-generation AI [142]. In 1979, Barone showed that the first steroidal AI testolactone could inhibit peripheral aromatization by up to 90% [143]. The first third-generation AI anastrozol was released in 1996 for the treatment of postmenopausal breast cancer patients and the largest study in the adjuvant setting, called ATAC study, was started. The superior efficacy of anastrozole compared to tamoxifen for the adjuvant therapy of HR-positive breast cancer in postmenopausal women was also demonstrated in the long-term follow up [144,145]. Today, non-steroidal AI anastrozole (Arimidex^®^) and letrozole (Femara^®^) and the steroid AI exemestane (Aromasin^®^) are available for clinical use. The following side effects are listed in the package leaflet and are similar to those of tamoxifen: depression is common in psychiatric disorders, occasionally anxiety (including nervousness) and irritability occur. In diseases of the nervous system, headaches, dizziness, less often, somnolence, insomnia, memory problems, sensory disturbances (including paraesthesia and hypoesthesia), taste disorders, cerebrovascular accident and carpal tunnel syndrome are common. Side effects of AIs include changes in mood, mental state, cognition and behavior. In the registration study of anastrozole in the ATAC trial, mood disorders were reported in 19.3% of patients receiving anastrozole versus 17.9% of patients receiving tamoxifen [144].

In mice models it could be demonstrated that letrozole is permeable and can cross the BBB to a high percentage in dose dependent manner [146]. In a further study in the mouse model, it could be shown that anastrozole and vorozole were transported by the P-gp, while neither compound was transported by BCRP. The authors propose that P-gp-mediated active efflux at the BBB limits the action of anastrozole in the CNS, while vorozole and letrozole easily cross the barrier [147].

There are several case reports and clinical data demonstrating that BM are stable on AI and patients benefit from such therapy regimens [148,149,150,151].

### 5.3. Fulvestrant

In 2004, fulvestrant, another anti-estrogenic substance (so-called SERD = Selective Estrogen Receptor Down Regulator) was approved [152,153,154]. In contrast to tamoxifen, this has no partial estrogen agonistic properties. Fulvestrant competitively binds to the ER with an affinity comparable to estradiol and completely blocks the trophic effects of estrogens [155].

The only neurological side effect listed in the pivotal study is frequent headache [153,156].

Fulvestrant does not appear to be able to cross the intact BBB. It is therefore assumed that neurological side effects occur less frequently. Overall, however, knowledge of its effectiveness in BM is limited [154,157].

### 5.4. GnRH-Analogue

Another important step in the therapy of ER-positive breast cancer was the development of the so-called GnRH analogues. The GnRH analogues (formerly LHRH analogues) are synthetic analogues of gonadotropin-releasing hormone (GnRH) and include GnRH agonists and antagonists. GnRH agonists cause the initial release of gonadotropins in the anterior pituitary gland. Thereafter, the down-regulation of pituitary GnRH receptors occurs followed by of the decrease of ovarian steroid hormone. Since the introduction of goserelin in 1990, the use of GnRH analogues as endocrine therapy for premenopausal breast cancer patients with ER-positive breast cancer has been approved [158,159,160].

The side effects of GnRH analogues are indirectly by suppressing LH/FSH in the anterior pituitary gland and thus as a subsequent reaction common estrogen-withdrawal symptoms, such as psychiatric disorders, including depression, sleep disorders and mood swings. Headaches and paresthesias, occasional dizziness and temporary changes in taste are often described in diseases of the CNS [161,162].

There is little basic research data addressing the uptake of GnRH analogues. However, these data confirm the uptake, which also explains the side effect profile of the drugs [163]. GnRh analogs are known to affect the endocrine system. Wilson et al. was also able to determine that reproductive hormones regulate the selective permeability of the BBB and thus changes in the hormonal situation can have an influence on the permeability of the BBB [164].

There are some clinical data and case reports of patients with cerebral metastases and therapy regimens with GnRH analogues [148]. A combination of GnRH analogues plus another endocrine therapy such as letrozol is commonly used in clinical practice.

### 5.5. Everolimus

In addition to the sole anti-hormonal therapy, based on the results of the BOLERO-2 study, the European Medicines Agency (EMA), approved everolimus for the indication “HR-positive, advanced breast cancer” at the end of July 2012 [165,166]. Everolimus is a selective inhibitor of the serine/threonine kinase mTOR (mammalian target of rapamycin), whose activity is upregulated in a variety of human tumors. Everolimus can be combined with AIs such as exemestan [165], letrozole [167] or tamoxifen [168] and fulvestrant [169].

Several side effects have been reported in clinical studies. For psychiatric disorders, insomnia has often been noted. In disorders of the nervous system, dysgeusia and headaches, and occasionally ageusia are reported very commonly. In eye diseases, eyelids edema and occasionally conjunctivitis were common side effects [165,166].

In mouse and rat models it could be demonstrated that everolimus can penetrate through the BBB [170,171].

In clinical studies, mTOR inhibitors such as everolimus and temsirolimus have been tested in recurrent globlastoma multiforme, but have not fulfilled the hope of a good tumor response [172,173].

As it is known that everolimus can cross the BBB, it was used in the LCCC 1025 study in HER2+ patients with BM in combination with trastuzumab, and vinorelbine as part of the therapeutic regimen [174]. Overall, the combination of everolimus, vinorelbines and trastuzumab did not reach the pre-defined intracranial response endpoint [174]. Another trial for HER2+ patients with BM involved the use of a combination of everolimus, lapatinib and capecitabine [175]. The results were promising, although only 19 patients were enrolled in the study.

There is no study investigating the use of everolimus in HR+ patients with BM. Even in real world data analyses, the significance of the effectiveness is limited [176]. The reason for this lack of information is that in clinical trials, the patients with a history of BM are excluded [165].

### 5.6. CDK 4/6 Inhibitors- Palbociclib/Ribocilib/Abemacilib

In HR-positive resistant metastatic breast cancer, inhibition of cyclin-dependent kinases 4 and 6 plays an important role. To date, three dual CDK4/6 inhibitors have been approved for the treatment of breast cancer. These were able to show that, in combination with endocrine therapy, they significantly improved survival results both in the initial and in the later therapy [177,178,179,180,181,182]. CDK 4/6 inhibitors can be combined with both an AI or fulvestrant [183,184,185,186,187,188,189,190].

#### 5.6.1. Palbociclib

Palbocilib has been approved in Europe for metastatic HR-positive breast cancer since November 2016 [183].

The side effects of palbociclib commonly reported are dysgeusia and frequently blurred vision, increased tear secretion and dry eyes [183,184,191,192].

Palbociclib is reported to be able to cross the BBB [193,194]. In in vivo animal studies, palbociclib was found to be a substrate of the efflux transporters P-gp and BCRP. However, it is precisely these transporters that impede the transport of palbociclib to the brain [195,196].

There was a study of palbociclib in patients with metastatic HER2+ breast cancer with BM (NCT02774681) [197]. This study was terminated due to slow accrual. In contrast only the PALOMA 2 trial included patients with BM (2 of 666 enrolled participants), in the other two admission studies patients with known uncontrolled or symptomatic BM were excluded [183,198].

#### 5.6.2. Ribociclib

Ribociclib was approved by the EU in September 2017 for combination therapy with AI [45,187].

Reported side effects are similar to the other CDK 4/6 inhibitors in nervous system disorders. Headache and drowsiness are very common, and dizziness is frequently reported. In eye diseases, increased lacrimation is mentioned, and dry eyes are common [187,192].

Ribocilib can cross the BBB and enter the brain [199,200]. Ribociclib is transported through the BBB by P-gp and is therefore also limited by this transporter [201]. There are case reports of patients with BM who benefit well from Ribociclib [202].

The MONALEESA-3 study in patients with advanced HR+ breast cancer, in which patients were treated with ribociclib in first- or second-line therapy, also included patients with treated and stable BM. A total of 8 patients with BM were included, 6 in the ribociclib group and 2 in the placebo group. However, an evaluation based on this small amount of data was not possible [188]. Another subgroup analysis of the phase IIIb CompLEEment-1 trial also supports the benefit of ribocilib in patients with BM [203].

#### 5.6.3. Abemaciclib

Abemaciclib was approved in September 2018 for women with HR+ breast cancer that is metastatic or locally advanced [189]. Compared to the other CDK4/6 inhibitors, abemaciclib has a higher lipophilicity and may therefore be able to penetrate breast tissue and the BBB more efficiently [204]. Abemaciclib can cross the BBB and could therefore have an effect on BM [205,206]. Additionally, preclinical data of abemaciclib in human xenograft models show reduced tumor growth in the brain. Abemaciclib was also shown to have the highest unbound brain to plasma ratio compared to palbociclib and ribociclib, indicating effective penetration [206].

A clinical study has shown that systemic treatment with abemaciclib leads to similar concentrations of abemaciclib in both plasma and cerebrospinal fluid [206].

### 5.7. Alpelisib

A mutation of PIK3CA is in 25–40% of ER+, HER2− breast cancers, and in 8% of ER- breast cancer [207,208].

Alpelisib in combination with fulvestrant has been approved in the EU since July 2020 for the treatment of patients with MBC and a molecular genetic mutation of PIK3CA. However, the pharmaceutical company Novartis announced in April 2021 that it would withdraw alpelisib from the German market because the price negotiations with the health insurance companies had not found an appropriate basis for a future reimbursement. Alpelisib is still approved in the EU, but now has to be imported from abroad [209].

Insomnia is a commonly reported side effects in psychiatric disorders, and headaches and dysgeusia are very commonly reported. Blurred vision and dry eye are often mentioned as eye diseases [210].

The pivotal study SOLAR-1 on alpelisib excluded patients with untreated or active CNS metastases [210].

Fitzgerald et al. analyzed in a retrospective cohort study that PIK3CA-activating mutations can be associated with an increased risk of CNS metastases in patients with ER+/HER2− disease [211].

In contrast, alpelisib does not appear to be able to cross the BBB in clinical use or in mouse models [212,213].

## 6. Conclusions and Future Perspectives

BM is less common in patients with HER2−negative hormone-sensitive metastatic breast cancer than in other molecular subtypes. Treatment options such as surgical removal or radiation can relieve symptoms but often have pronounced side effects, standing in contrast to the actual prolongation of survival through these treatments is questionably convincing in clinical data.

It is therefore important to have available anti-hormonal therapeutics that can cross the BBB and thus act intracerebrally. Not all common anti-hormonal therapeutics are able to penetrate the CNS. It is therefore important for the treating oncologists to use substances that have been proven to cross the BBB, despite the limited data available.

AIs, especially letrozole, probably also tamoxifen, everolimus and CDK4/6 inhibitors, especially abemaciclib, appear to act intracerebrally by overcoming the BBB.

Despite everything, there is only limited clinical data, but also from basic research on anti-hormonally active substances and cerebral metastasis of HR+ HER2− breast cancer.

It is positive that, at least recently, there have been subgroup analyzes in approval studies that take patients with BM into account.

Nevertheless, further data must be obtained in the area of basic research, but also health care research in relation to patients with BM.

## Figures and Tables

**Table 1 cancers-14-05132-t001:** Summary of antihormonal therapies and their blood-brain barrier permeability.

Antihormonal Therapy	Blood-Brain Barrier Permeability
Tamoxifen	positive
AI	positive
Fulvestrant	negative
GnRH-analogue	positive
Everolimus	positive
CDK 4/6 inhibitors	
Palbocolib	positive
Ribociclib	positive
Abemaciclib	positive
Alpelisib	negative

## Data Availability

Not applicable.

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
