# Peer review of "Anti-Hormonal Therapy in Breast Cancer and Its Effect on the Blood-Brain Barrier"

_cancers, 2022, doi:10.3390/cancers14205132_

Round 1
Reviewer 1 Report
Dear authors,
thank you for the review to this relevant clinical topic. The review provides a good overview over the endocrine therapy of breast cancer incl. the mechanism of action as well as evidence of activity of the endocrine therapy in the brain.
I suggest following minor changes to optimize your manuscript:
1. I advice to use the terminology luminal A-like instead of luminal A and luminal B-like instead of luminal B as in the performed context no gene expression analysis were made.
2. I suggest to refer the international guidelines (for example ESMO Guidelines) in addition to AGO.
3. Please check if there is evidence that brain surgery doesn't improve survival in patients with brain metastases of breast cancer . You write "Treatment options such as surgical removal or radiation can relieve symptoms but do not prolong life and often have pronounced side effects". To my knowledge there is no qualitatively good published data comparing for example brain surgery vs. no surgery in patients with brain metastases of breast cancer.
3. I suggest to add the aspect of patients with triple-positive breast cancer with brain metastases as this subgroup has the best prognosis among patients with brain metastases of breast cancer and further analysis are urgently needed to evaluate the role of modern endocrine treatment concepts ( for example in comparison mit HER2-targeted agents) for this patients cohort.
Author Response
Dear reviewer,
Thank you for your comments. We have adapted our script to your following points.
- The terminology was adapted in line with your suggestion
- In addition to the AGO recommendations, we have included the international guidelines of the ESMO guidelines
- We included the general concern about the lack of good quality of published data regarding brain surgery versus no surgery in patients with recorded brain metastases from breast cancer (see 3.3. and Conclusion)
- We added a section on triple-positive breast cancer and BM (see 4.4.)
Thank you so much for your time reviewing my script.

Reviewer 2 Report
The authors (Curtaz, et.al.) of this manuscript entitled ‘Anti-hormonal therapy in breast cancer and its effect the blood-brain barrier’ describe the effect of specific medications for breast cancer metastasis in the brain. Overall, the manuscript is well-written and easy to read. The authors are only encouraged to pay attention to the following minor comments:
1. The title is missing a word ‘Anti-hormonal therapy in breast cancer and its effect on the blood-brain barrier’.
2. It would be very interesting to discuss in a small section if breast cancer cells metastasizing to the brain still retain the ability to respond to the same anti-hormonal drugs as the original tumor -if there are any direct experimental evidence.
3. Also, it would be critical to discuss if and how cancer cells alter the blood-brain barrier and therefore affect the ability of drugs to reach the brain.
Author Response
Dear reviewer,
Thank you for your comments. We have adapted our script to your following points.
- Thank you so much for this comment. We have corrected the title accordingliy.
- We have added a section in the chapter 3.2., which discusses this very important point. On the whole, however, there is no well-founded basic research on this.
- We have added a section in the chapter 3.2., which discusses this very important point.
Thank you so much for your time reviewing our script.
